# A Survey on Techniques in the Circular Formation of Multi-Agent Systems

**Hamida Litimein [1]** **, Zhen-You Huang [1] and Ameer Hamza [2],***

1    Department of Applied Mathematics, School of Science, Nanjing University of Science and Technology, Nanjing 210094, China; litimein.hamida@outlook.com (H.L.); zyhuangh@njust.edu.cn (Z.-Y.H.)
2    Department of Electrical and Computer Engineering, COMSATS University Islamabad, Lahore 54000, Pakistan
*    Correspondence: ameerhamza@cuilahore.edu.pk

**Abstract:** Distributed control solutions to the multi-agent systems in terms of coordination, formation, and consensus problems are extensively studied in the literature. The circular formation control of multi-agent systems becomes one of the most challenging and active research topics in distributed cooperative control. The research potential of circular formation control has grown especially in satellite formation and control. The circular formation control aims to drive a group of agents to move on a circular trajectory around a center with spacing adjustment to avoid the collision. This paper covers the fundamental research in consensus and formation control and the latest developments in the applications of formation control especially in circular formation control and stability analysis. Literature related to various control streams to achieve circular formation is discussed. Algebraic graph theory is used as a building block in problem definition, problem formulation, and solutions. A review of bearing, range, and received signal strength measurements with respect to circular formation control is presented. Leader following, behavior-based, artificial potential function, virtual structure, and cyclic pursuit are the methods discussed most in this context. The stabilization of the collective circular motion with the unicycles approach is also discussed. Further results on circular formation, several applications, and future research directions are presented.

**Keywords:** circular formation; collective circular motion; cycle pursuit; multi-agent systems

## 1. Introduction

Distributed formation control in multi-agent systems (MASs) has attracted tremendous attention during the last decade due to its wide applications in the real-world including surveillance [1], distributed sensor networks [2], environmental monitoring [3], and pursuit and evasion [4]. One of the fundamental problems in distributed formation-based MASs is to achieve consensus or agreement. The consensus-based formation control has been studied in [5,6]. Circumnavigation problems for target tracking [7] and cyclic pursuit [8] have been proposed to consider different cooperative tasks such as coordination in sensor networks [9], satellite formation, and control [10] in MASs. The introduction of the coordination architecture including leader–follower, behavioral, and virtual structure approaches is presented in [11]. In [12], an overview of formation control with reference to position-based control, displacement-based control, and distance-based control is presented.

Among the important problems proposed in formation control, the circular formation problem (CFP) has received much attention from the research community. Many research works in the circular formation control have been carried out due to its intensive applicability in many applications such as entrapping a target [13], patrolling and surveillance missions [14,15], and source-seeking in a sensed environment [16]. Generally, circular formation control is formation control where the interactions between agents are characterized by cycle/directed cycle graphs (with possibly the addition of a center node). The results in CFP have been divided into two categories. The first category considers formation control

with cycle/directed cycle and is also referred to as collective circular motion (CCM) [17–19]. The second category considers formation control with cycle/directed cycle with an additional center node is also referred to as circumnavigation [20,21]. A collective circular motion stabilization is studied in [22,23] with a focus on many practical issues where the agents are required to perform a collective motion such as sampling of ocean variables using a fleet of self-directed underwater gliders [24]. Applications of circumnavigation in MAS applications comprise of protection of VIPs, surveillance, and target localization where the agents may need to circulate in different orbits to perform different missions around the target for monitoring or gathering information [25,26]. The objective of all the above applications is to achieve circular formation.

CFP of MASs using first-order integrator dynamics is discussed in [27,28]. CFP for double integrator dynamics is presented in [29–32]. The circular formation control for the unicycle model is proposed in [33–35]. Most of the studies on CFP addressed the evenly spaced circular formation where the spacing among the agents is fixed and equal [36,37]. However, in certain applications such as collecting the maximum information of the target and preventing an enclosed target from escaping [38], the agent does not need to be evenly spaced on the circular trajectory. Moreover, the evenly spaced distribution may not guarantee a good outcome in unknown situations. Thus, to address this problem, the CFP with arbitrary spacing is desirable. Achieving a circular formation with arbitrary angular spacing is a challenging problem. Only a few research studies addressed the problem of the arbitrary spacing distribution in a circular formation [27,39–41]. In an arbitrary distribution, the main challenge is to design a control structure that has control over the inter-agents arbitrary spacing around the target. In [39,40], distributed control protocols are designed to form a circular formation with arbitrary spacing. However, the agents are initially located on the circular trajectory and move only on one-dimensional space. The authors in [27] designed a control law based on limit-cycle oscillators, which have a control structure over the arbitrary spacing of agents in a circular formation in 2D space. In [41], an estimator is provided to ensure any arbitrary inter-agent angular spacing.

The symmetric phase pattern referred to as the synchronized and the balanced circular formation has been investigated in [42,43]. Synchronization means the case when each agent moves in the same direction in time with other agents [42]. Balanced circular formation means the case when all agents move with respect to a fixed center, which represents the average position of the agents [44,45]. The stabilization of agents in the collective circular motion around a common circle under the assumption that each agent moves initially around an individual circle with different radii and different centers is presented in [46]. The results have been extended to CCM stabilization with phase arrangements using second-order rotational dynamics in [42]. A fail-safe strategy based only on the bearing measurements is proposed using unicycle kinematics to achieve the balanced circular formation where the center may or may not be pre-specified [47].

Due to the extensive literature on circular formation control, it will be very important to review the existing results on CFP and provide an overview of the latest progress in circular formation control, which can provide novel research directions. We focus on the classification and formulation of the general CFP by providing the various methods and measurements to solve the problem. A detailed analysis of CFP with relevance to measurements of agent's data such as position [48], distance [49,50], displacement [51,52], bearing (angel/position) [53,54], range [55], and received signal strength (RSS) [2] measurements opened new research paradigms. Presentation of a method-based classification of CFP, such as behavior-based method [56–58], leader–follower method [59,60], virtual structure method [61], artificial potential method [62–64], and the cyclic pursuit method [65–67] using the algebraic graph topology are the main context of this paper. The solution of CFP under different communication constraints such as the input saturation [68,69], disturbances in system dynamics [70], and locomotion constraint [40] are of extreme significance with respect to practical applications. The CFP has been addressed via source seeking [71] where successful attempts are made to find an unknown source/target. Coverage con-

trol [72] to cover/navigate a particular circular space and event-triggered control [73,74] where a particular event in CFP triggers the control/stability phenomena.

This paper aims to present a comprehensive review of the theoretical and practical advancements in circular formation control of MASs. Control methodologies, applications, and future directions represent different aspects of CFP. Several existing survey articles discuss only the formation control of MASs. The novelty of this paper is, it is the only review paper that discusses the circular formation problem in MASs. Furthermore, it also covers the existing and recent development and progress in the circular formation. The motivation for this review article is to identify the control challenges in the circular formation of MASs.

The rest of the paper is organized as follows: the generalized problem formulation and the classification of the circular formation control are presented in Section 2. Section 3 discusses the control methods in the circular formation problem. The stabilization of the collective circular motions is discussed in Section 4. In Section 5, event-triggered circle formation control is discussed. Circular formation control with constraints is presented in Section 6. In Section 7, source seeking via circular formation is addressed. Applications of circular formation are discussed in Section 8. The future research directions are discussed in Section 9. Finally, conclusions are given in Section 10.

## 2. Formulation and Classification of the CFP

### 2.1. General Circular Formation Control Problem

Circular formation control is to lead a group of agents to orbit around a center on a prescribed circular trajectory with spacing between the neighboring agents. The CFP can be formulated as a general model setup for $n$-agents as follows:

$$
\begin{aligned}
\dot{x}_i &= F_i(x_i, u_i), \\
\dot{y}_i &= G_i(x_1, x_2, \ldots, x_n), \\
\dot{z}_i &= H_i(x_i), \quad i = 1, 2, \ldots, n,
\end{aligned}
\tag{1}
$$

where $x_i \in \mathbf{R}^{n_i}$, $u_i \in \mathbf{R}^{m_i}$, $y_i \in \mathbf{R}^{p_i}$, and $z_i \in \mathbf{R}^q$ are, respectively, the state, control input, measurements, and the output. The functions are defined as follows $F_i : \mathbf{R}^{n_i} \times \mathbf{R}^{m_i} \to \mathbf{R}^{n_i}$, $G_i : \mathbf{R}^{n_1} \times \mathbf{R}^{n_2} \times \ldots \times \mathbf{R}^{n_n} \to \mathbf{R}^{p_i}$, and $H_i : \mathbf{R}^{n_i} \to \mathbf{R}^q$. Table 1 lists the notations used in this manuscript.

Under this setup, the general CFP can be stated as follows: Consider the model (1). Design a distributed control law $u_i$, $i = 1, \ldots, n$, to lead a group of agents to spread and move on a common circular trajectory. The center is specified or determined by the initial states of the agents. The initial states depend on the common agreed sense (i.e., clockwise or counterclockwise) and spacing between the agents to avoid a collision.

The output of the system will converge to a consensus or rendezvous problem. The output for the model (1) also converges the error dynamics to zero based on position-based formation control. The extension of the model (1) may also be applied to solve containment control problems by converging the follower agent's error to zero based on the leader's position, displacement, and angle.

**Table 1.** Notations used in this paper.

| Notations | Definitions |
| --- | --- |
| $\mathbf{R}^n$ | The $n$-dimensional Euclidean space |
| $\mathbf{R}^+$ | The set of positive real numbers |
| $\mathbf{N}$ | The set of non-negative integers |
| $\mathbf{C}$ | The set of complex numbers |
| $A^T$ | The transpose of $A$ |
| $\|\cdot\|$ | The Euclidean norm |
| $I$ | $2 \times 2$ identity matrix |
| $sgn^+(x)$ | $sgn^+(x) = 1$ if $x \geq 0$ otherwise $sgn^+(x) = 0$ |
| $< z_1, z_2 >$ | The scalar product of $< z_1, z_2 > = Re(\bar{z_1}^T z_2)$ for $z_1, z_2 \in \mathbf{C}^N$ |
| Class $\mathcal{S}$ | $\mathcal{S}(x) = \{f : [0, \infty) \longrightarrow [0, a] \mid f$ is Lipschitz continuous and $f(\tau) = 0$ if $\tau \geq x$ otherwise $f(\tau) > 0$, $0 < a < \infty\}$ |
| $\mathcal{G}$ | A weighted digraph $\mathcal{G} = \{\mathcal{V}, \mathcal{E}, \mathcal{A}\}$ where $\mathcal{V} = \{\mathbf{v_1}, \cdots, \mathbf{v_n}\}$ represents a set of vertices, $\mathcal{E} \subseteq \mathcal{V} \times \mathcal{V}$ represents the set of edges, and $\mathcal{A} = [a_{ij}]_{n \times n} \in \mathbf{R}^{n \times n}$ is the weighted adjacency matrix, for an edge $(i, j) \in \mathcal{E}$, $i$ denotes an in-neighbor of $j$ and $j$ denotes an out-neighbor of $i$ |
| $\mathcal{N}_j^{out}$ | The out-neighbors set for given node $j$ |

### 2.2. Classification of the CFP

The circular formation can be simply divided into two categories, CCFP and DCFP. The categorical division is made with respect to the sensed information (global, local). The DCFP is composed of distributed circular formation [75] and hierarchical circular formation [76]. Most of the existing results in the literature study the decentralized formation control schemes due to the high reliability and the good flexibility compared to the centralized formation on which the reliability is poor [12]. The CFP could be static or dynamic. In the static case, the radius of the circular trajectory, angular speed, and spacing between the agents are assumed static [77]. If at least one of the quantities varies (speed, position, spacing, etc.), the problem will be considered as a dynamic CFP [77–79]. The characteristics of CFP with reference to control and stability are further classified with respect to the center as follows [80]:

1. Formation control with cycle/directed cycle: It is also referred to as CCM. It represents a circular formation with an unspecified center. In this case, the center of the circular formation is determined by the initial states of the agents.
2. Formation control with cycle/directed cycle with an additional center node: It is often referred to as circumnavigation. It represents the circular formation with a given center. Circumnavigation or target-enclosing refers to encircling around a specific target by a single agent or group of agents.

Research work by different authors related to CFP and control design has addressed the circumnavigation problem since the center can be considered as a target or a reference beacon [80,81].

The CFP is studied such that the agents are evenly or arbitrary distributed on the circular trajectory [27]:

1. Even distribution: The circular formation is achieved such that the agents are evenly distributed as shown in Figure 1a and equally spaced on the circumference of the circle, this problem has been addressed in [17,82]. A distributed control law for evenly spaced multiple mobile robots to achieve circular formation is proposed in [83]. Achieving even distribution in circular trajectory by autonomous mobile robots while enclosing a moving target is presented in [84]. A hybrid control law based on local information is designed for a group of unicycles to move in uniform collective motion around a target based on even distribution [85]. The even distributions describe many practical applications but it may not ensure the optimal configuration [68,79].
2. Arbitrary distribution: Any arbitrary or desired distribution on a circular path is shown in Figure 1b. The circular formation is achieved such that the agents are

arbitrarily distributed on the circle in [27,86]. Distributed control algorithms are designed first to lead a group of agents to move on a circular path while preserving their spatial order [39]. An extension based on sampled data and finite time control was proposed to make the problem more application-oriented [39]. The circular formation for multiple agents can be achieved with any preset phase distribution by combining the circular motion control and phase control in the two-dimensional space [87]. The static and dynamic circular formation control is studied in [77,88] for a heterogeneous multi-robot system with arbitrary spacing. Arbitrary distribution encourages the control design for collision avoidance as compared to even distribution. This makes arbitrary distribution-based CFP more application-oriented.

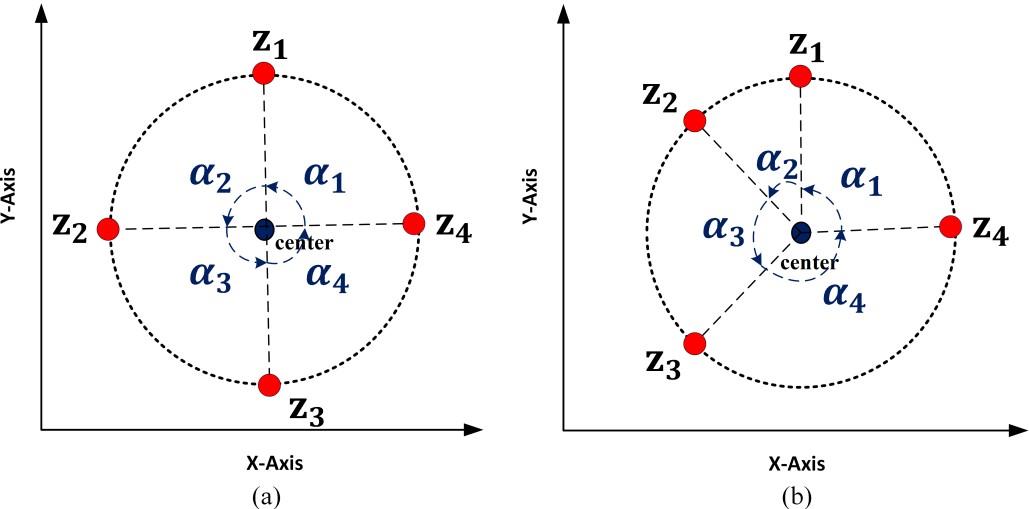

**Figure 1.** Circular formation of four agents in a plane: (**a**) evenly spaced circular formation with equal angular separation $\alpha_i = \frac{2\pi}{n}$; (**b**) arbitrary spaced circular formation with unequal angular separation $\alpha_i \neq \frac{2\pi}{n}$.

Based on the aforementioned observations, the CFP methods are classified as: Artificial potential method [89,90], cyclic pursuit method [91,92], leader–follower method [93], behavior-based method [94,95], and virtual structure method [96,97]. The comparison of mentioned methodologies is summarized in Table 2. Note that above methods can be used together or separately to solve the CFP [75,98].

### 2.3. Categorical Measurements in a Circular Formation

The measurement approaches for CFP are studied using classifications such as position, displacement, distance, bearing (angle/position), range, and RSS measurements. The distinctions among the position, displacement, and distance measurements based formation control in terms of the sensing capability and the interaction topology are discussed in detail in [12]. The distinction between the bearing and range measurements is presented in Figure 2. The classification of the above-mentioned schemes related to measurements in CFP could be amalgamated together to solve the CFP [99].

#### 2.3.1. Bearing Measurements

The bearing angle refers to the angle subtended at agent *a* by its two neighbor agents [100]. Figure 2a illustrates the bearing measurements where $\theta_{ab}$ denotes the bearing angle between the velocity vector of agent *a* (along with body *x*-axis) and vector $d_{ab}$ which connects the two agents *a* and *b*, $\theta_{ao}$ denotes the bearing angle between the velocity vector of agent *a* (along with body *x*-axis) and vector $d_{ao}$ which connects the agent *a* and the center *o*, $\phi_{ab}$ denotes the separation angle between agent *a* and *b* with respect to the center *o*, and $\theta_a, \theta_b$ denotes, respectively, heading angles of agents *a* and *b* with respect to *x*-axis. The inter-agent angles can also be used and controlled to achieve the desired

formation [12]. Under bearing measurements, the desired formation is defined by inter-neighbor bearings [100]. The advantage of measuring the bearing angle in solving the CFP reduced invariance in rigid body transformation with respect to invariant scaling [101]. The bearing measurement is defined as a passive measurement technique [102]. Practical applications such as surveillance and navigation require control schemes based on bearing measurements are discussed in [49,54]. In some scenarios such as tracking, the application of the bearing measurements necessitate some observability conditions [103]. CFP by using bearing-only measurements is studied in [45,47,104]. Different from the bearing measurements, the bearing-only measurements require the agents to calculate only the bearing angle without any relative position measurements [101]. The bearing-only measurements require sensors such as the vision sensors, for example, monocular or omnidirectional cameras, for measuring the bearing angle efficiently [105]. Other types of sensors such as the passive radars and sensors arrays are also used to calculate the bearing measurements [105]. The finite-time stabilization of angle-constrained circular formation is studied in [106] where each agent can only measure the bearing of their neighbor agents. Circular formation control based on the bearing measurements is investigated in [53] for a group of kinematic unicycles with arbitrary phase allocation. A cost estimator for target localization and circumnavigation is discussed in [107]. Note that the bearing-based CFP is solved as a linear control problem whereas the bearing-only CFP is solved as a nonlinear control problem [105]. It is worth mentioning that in many research works the bearing vectors or line-of-sight are more general and are being used instead of angles [108,109]. Distributed cooperative control design for cyclic formations based on bearing-only measurements is presented in [108]. The surrounding formation control problem for star framework by using bearing-only measurements is investigated in [109].

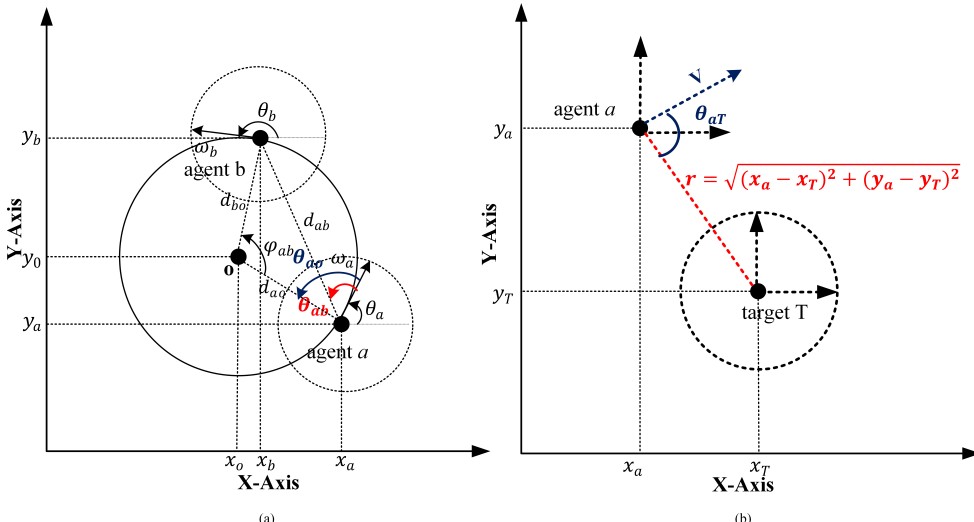

**Figure 2.** Circular formation measurements setup: (**a**) bearing measurement setup; (**b**) range measurement setup.

**Table 2.** Comparison of circular formation control methods.

| Methods | Definitions | Advantages | Disadvantages |
|---|---|---|---|
| Cyclic Pursuit Method [17,20,110–113] | • For $n$ agents, each agent $i$ follows its next agent $i+1$ modulo $n$. | • A distributed control method does not require a leader (pursuer) to achieve the formation that guarantees the anti-interference/anti-disturb performance.<br>• It only necessitates relative measurement that includes only position and speed. Thus, it can reduce the communication interaction between the agents by using a minimum number of communication links to achieve the desired formation and retain the connectivity ($n$ links for $n$ agents).<br>• It can keep the agent's formation in collective uniform configuration and reorientation.<br>• In this method, all agents converge toward one point (center). Thus, it ensures the capturability of the target. | • It introduces feedback paths, which makes stability analysis difficult. |
| Leader-follower Method [27,30,35,36,41,50,60,75,84,85,93,114] | • At least one agent is designated as the leader and the other agents are considered as followers. The follower agents track the leader such that no feedback exists from the follower agents to the leader. | • Intuitiveness, simplicity, and ease implementation of using decentralized approaches where every agent takes its own decision based on local information.<br>• The desired formation can be specified in terms of the leader's trajectory and it can still maintain even if the leader's motion is subject to some disturbances. | • The leader's agent is highly independent of the follower agents, i.e, lack of feedback from the follower agents to the leader agent.<br>• Formation does not allow the leader's faults. If the communication between the leader and followers is interrupted, it is difficult to maintain the formation. |
| Behavior-based Method [33,45,57,81,94,95,98,115] | • Multiple expected and desired behavior is prescribed for every agent such as collision avoidance, obstacle avoidance, formation reconfiguration, target tracking, etc. | • Decentralized control method where the group dynamics incorporates the formation feedback through neighbors-based communication.<br>• High adaptability to systems with multiple interactions, particularly for large-scale robots. | • The overall behavior cannot be explicitly defined and the mathematical analysis of this method is difficult. Thus, the convergence to the desired configuration cannot be assured. |

**Table 2.** *Cont.*

| Methods | Definitions | Advantages | Disadvantages |
|---|---|---|---|
| Artificial Potential Method [21,63,64,89,90,116–118] | • This method constitutes attractive and repulsive forces, which generate the potential function. | • The simplicity and robustness of real-time control make it suitable for real-time applications and good performance in dynamic environments.<br>• Distributed control strategy that is applicable for controlling large systems/groups of robots.<br>• The agents are capable of dealing with different desired/expected tasks.<br>• Improve scalability of agents formations, obstacle/collision avoidance, and reconfiguration. | • Producing the local minima problem in which the agent is trapped before achieving the desired goal.<br>• Difficult to design the local basic behavior and control of the desired formation. |
| Virtual Structure Method [61,96,97,119,120] | • The formation of the agents is treated as a rigid body structure based on a virtual reference point or virtual leader. | • This approach can easily specify the coordinated behavior for the group without an explicit leader, and that the desired formation can be highly maintained during the maneuvers.<br>• It does not rely on a single real unit which makes it highly robust than the leader–follower approach.<br>• It has a simple formation definition since the agent follows the center of the body instead of another agent. | • It is a centralized method where the central control unit manages the formation keeping. Thus, it increases time consumption due to the computational prediction of the system in finite time. |

2.3.2. Range Measurements

Under the range measurements, each agent can measure the ranges, which are defined by the distances to their neighbors only [121]. It is used in the case where limited information is available [122]. In this case, it is required to develop control schemes based on the range and range rate (i.e., the derivative of range) measurements. Recently, the design of control law based on the range measurement where the range and the range rate are available becomes a challenging research topic in circular formation control [121]. Under this measurement, the sensors provide only the relative distance between the agents and the target as illustrated in Figure 2b where $r$ denotes the range between the agent $a$ and target $T$, $\theta_{aT}$ denotes the bearing angle of agent $a$ with respect to the target $T$, and $V$ denotes the constant velocity of agent $a$. The range and the range rate measurement have been used in [121–123], the range is given by the Euclidean distance $r = \sqrt{((x - x_T)^2 + (y - y_T)^2)}$, where $(x, y) \in \mathbf{R}^2$ represents the agent position and $(x_T, y_T) \in \mathbf{R}^2$ is the target position. The guidance law was designed such that the distance from the agent to the target $r(t)$ converges to the desired radius $r_d$ in finite-time based on estimation of the target location [123]. It was mentioned that the rate measurements can be replaced by an estimated range rate, which is estimated by sliding-mode estimator [123]. Cooperative tracking based on the range-only control to achieve circumnavigation in unmanned aerial vehicles is discussed in [124]. In the case where the GPS is not available, the range and range rate measurements can be used without localizing the target location to guide the agents to the desired circular trajectory [125].

The received signal strength (RSS) also gives range/distance information. The RSS measurements are usually a passive location estimation method. Under this measurement, each agent is equipped with sensors, which measure the signal strength of the received signal such that the distance to the target is estimated by the radio propagation model [56]. However, it is suggested to avoid using the distance measurements in the case where the agents are required to keep a radio silence such that the position shall not be detected [56]. The proactive sensors and range-free methods are more conservative and less costly compared to the range and bearing measurements [2]. The disadvantage of this measurement appears in the connectivity loss [126]. The RSS measurements can be also used in circular formation due to its applications such as the location estimation [2]. A distributed control scheme based on the energy intensity field is developed to achieve the multiple targets localization and pursue a group of mobile robotics sensor networks [2].

**Remark 1.** *In most practical applications, it is impossible to obtain the position or distance measurements. To address those applications, the circular formation has been studied for holonomic single integrator dynamics based on bearing-only measurements in [56]. Recently, a static control algorithm is designed without distance measurements and under velocity constraint for nonholonomic agents to maintain the spaced circular formation [80]. It was proved that the desired spaced circular formation is achieved with limited information about the center [80].*

## 3. CFP Methods

### 3.1. Circular Formation Based Cyclic Pursuit Strategy

Based on the graph theory of MASs, the formation control approaches can be divided into two different classes, cyclic and acyclic [127]. Acyclic topology is similar to the usual leader–follower technique [128] while in cyclic topology every agent follows other agents [110]. The early works on cyclic pursuit strategy are presented in [129,130]. In distributed cooperative control, the cyclic pursuit is considered as a swarm behavior [131]. Under cyclic pursuit approach, for $n$ number of agents, each agent $i$ follows the next agent $i + 1$ modulo $n$ [110]. This method requires a directed cycle graph to present the communication topology among agents as shown in Figure 3. The cyclic pursuit models have many practical applications in cooperative control and swarm robotics including ground robots [132], unmanned aerial vehicles (UAVs) [133], and satellites [134]. Cyclic pursuit is

advantageous since it requires less number of communication links, and drives all agents towards one point [131], which makes it a more application-oriented and attractive choice.

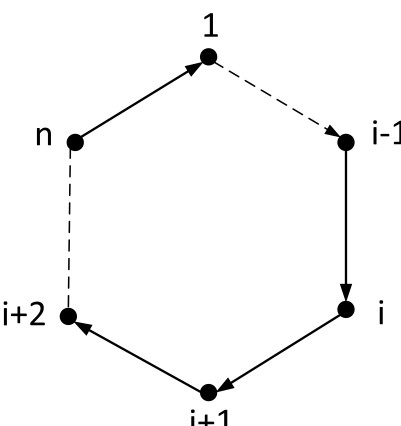

**Figure 3.** Cyclic pursuit scheme.

The cyclic pursuit strategy has been applied in the formation control of MASs using local control strategies. In [111], the regular polygon formations are studied in the plane for $n$ mobile agents, each agent starts with an arbitrary initial condition. The agents are modeled as first integrator kinematics:

$$\dot{z}_i = u_i, \quad i = 1, \ldots, n, \tag{2}$$

where $z_i \in \mathbf{R}^2$ and $u_i \in \mathbf{R}^2$ are, respectively, the position and the control input of the agent $i$. The linear cyclic pursuit control input was proposed as follows:

$$u_i = k(z_{i+1} - z_i), \quad i = 1, \ldots, n, \tag{3}$$

for $k > 0$. The system (2) and (3) can be expressed by the following linear system:

$$\dot{z} = Mz, \tag{4}$$

where $z = [z_1, z_2, \ldots, z_n]^T \in \mathbf{R}^n$, $M = circ([-k, k, \ldots, 0]) \in \mathbf{R}^{n \times n}$ represents the circulant matrix. The characteristic polynomial of matrix $M$ is $p_M(\lambda) = (\lambda + k)^n - k^n$. The matrix $M$ has a zero eigenvalue and all others have negative real parts. Thus, based on [130], for any initial position the center of the agents $z_1(t), z_2(t), \ldots, z_n(t)$ at time $t \geq 0$ remains unchanged and each agent $z_i(t)$ exponentially converge to its center. It was shown that the agents perform a regular pattern formations around the centroid, which is calculated from consensus problem [111]. According to [135], the Laplacian matrix associated with the graph representing cyclic pursuit is $L = -\frac{1}{k}M$ for $k > 0$. Note that the spectral properties of Laplacian are very important to compute the performance and robustness of dynamical systems. The stability analysis of the system (2) is determined by the location of the eigenvalues of the Laplacian matrix $L$ [5]. A generalization of the linear cyclic pursuit strategy with heterogeneous gains is studied in [65] to lead a group of agents to rendezvous at some desired point. The authors showed that by choosing proper gains, the agents could achieve rendezvous at some desired point. Further, by selecting proper deviations on agents [136] expands the reachability set presented in [65] to include points that are not reachable using generalized cyclic pursuit strategy. The necessary and sufficient conditions on the consensus of heterogeneous cyclic pursuit strategy in discrete time are established in [137]. Moreover, the results in [111] are extended to unicycle kinematics with identical constant velocity in [17] and with nonidentical constant velocity in [138]. Cyclic-pursuit control laws for both first and second-order integrator dynamics are designed for symmetric formations including the circular formation in three dimensions [112]; therefore, depending

on the angular velocity on the number of agents limits its applications. Furthermore, the circular cyclic pursuit model for agents modeled by the double integrator dynamics was studied in [139]. It was shown that under the proposed control model the system achieves the global configuration where both the agent speeds and the inter-agent distances converge to a constant value. In this technique, the pursuit angular velocity is not controlled [139]; thereafter, those limitations have been removed in [140] where the pursuit direction and angular speed are independent of the number of agents.

Recently, a modified cyclic pursuit strategy called deviated cyclic pursuit is introduced in [141] for a group of kinematic unicycles to achieve the balanced circular formation. A distributed control protocol of multiple unicycles is designed under ring topology such that the center is known only to one unicycle [20]. It was proved that the proposed control protocol leads the agents to a prespecified circular formation by taking into account the velocity constraint. A distributed control strategy based on state-space decomposition and complex matrix methods have been applied to solve CFP using first order and second-order integrator dynamics [113]. Based on the vector field approach, a regular polygon formation control strategy is achieved for double integrator MAS [142]. The feature of the proposed approach is that it allows the arbitrary control of the swarm centroid and angular velocity. Based on the cyclic pursuit scheme, a control law is developed for a group of multiple unicycles to orbit around a moving target unicycle in a common evenly spaced circular formation [143]. In order to solve the problem of moving target and hunting for multiple AUVs, a sliding mode formation control based on cyclic pursuit is designed in [144]. Cyclic pursuit in the presence of noisy proximity measurements is studied in [145] in which a novel estimation control strategy is designed to achieve a balance formation on the circle. Moreover, the authors in [146] discussed formation control on a closed curve, this technique may be used to solve the circular formation problem on a closed circular curve under an intermittent measurement flow. Cyclic pursuit-based strategy was presented in [92] to achieve the distributed cooperative moving-target capturing in 3D space. Further, [66] proposed a novel cooperative control technique based on cyclic pursuit strategy combined with fuzzy PD control to accomplish the circular formation configuration of MAS in 3D space.

*3.2. Target Tracking Circular Formation Control*

This section discusses leader–follower MASs topology with single or multiple leaders. The leader is considered as the fixed and static or dynamic and moving target point [30,84,85]. The target acts as the center of the circular trajectory around which the agents move or circular trajectory with fixed or transient radius [41,75,114]. Moreover, all the follower agents are required to maintain a desired equal distance to the leader to achieve the desired circular formation [27].

3.2.1. Fixed Target Tracking

In this framework, a leader–follower circular formation has been proposed. Consider a group of $n$ anonymous agents modeled as (2) and initially located in the plane with a leader or target position $z_0 = [x_0, y_0]^T \in \mathbf{R}^2$ as shown in Figure 4a. The objective is to drive the agents to follow a circular path with the dynamics of the target as the center of the circular trajectory as shown in Figure 4b.

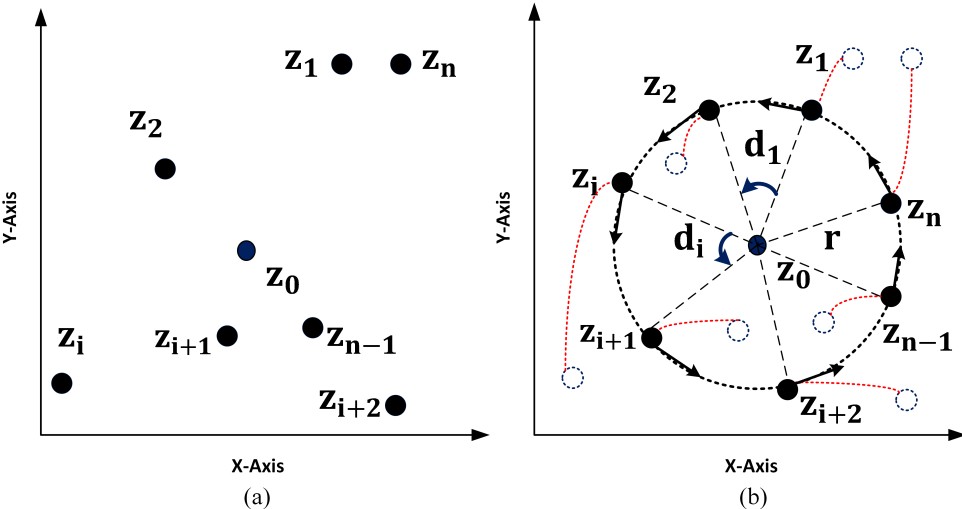

**Figure 4.** Circular formation with fixed target: (**a**) agents are initially positioned in the plane; (**b**) circular formation around fixed target.

To solve this problem, a limit-cycle decoupled design approach was proposed by using the relative position information, the controller is designed as follows [27]:

$$u_i(t) = u_i^p(t) h_i(t), \tag{5}$$

for $i = 1, \ldots, n,$. The simplification of control law (5) is presented in Appendix A. In this technique, the first part $u_i^p$ is designed for target circling where the agents are required to converge to circular formation in 2D space and rotate around the target in a counterclockwise direction with the desired radius determined as $(||z_i - z_0||_2 = r)$. The second part $h_i$ is designed for spacing adjustment between every agent and its neighbors characterized by desired distribution vector $d = [d_1, d_2, \cdots, d_n]^T$ [27]. Based on the leader–follower method and potential functions, distributed control protocols are designed to achieve the circular formation around the leader with desired radius and spacing between the follower agents [75]. The extension to double integrator dynamics is presented in [30]. In the case where the target position is unknown, a distributed control law is proposed to surround a convex target by a group of agents under the switching graph topology [147]. A position estimator as well as the control law is required to localize the target position. Based on distance measurements, an algorithm is proposed in [49], which executes two steps the localization step and the control step to achieve the circumnavigation of the unknown target by a single agent. It was concluded that the stability is established for both the stationary and slow drift case by using only the distance measurements to the target. Moreover, a similar problem has been studied by using the bearing-only measurements in [56,102]. Later, the results of localization and circumnavigation problem for multiple agents around an unknown stationary target is extended in [148] for 3D space. Recently, by using bearing measurements, a distributed control algorithm is designed to enable a group of agents to localize and circumnavigate a prescribed target [149].

For nonholonomic robots, various control algorithms have been proposed to achieve the circular formation tracking around a stationary target [20,85]. Control algorithms are designed for a group of nonholonomic vehicles to achieve CCM based on a fixed virtual reference beacon [33,81]. The circular formation stabilization is achieved for a group of dynamic unicycles where the topology of the unicycles contains a spanning tree [35]. It is shown that the model can be reduced to the kinematic unicycle case. The proposed control law for the unicycle model achieves circular formation stabilization around a given center. The information is known only to one unicycle subject to a given velocity constraint. A contraction control technique is applied to achieve the uniform distribution of agents along the circular trajectory for a class of nonlinear MASs at a

fixed center [150]. A time-varying radius reference and limited communication range is considered [150]. By using the local bearing measurements, a control scheme is designed to steer the agents toward circular formation around a static target [36]. Based on nearest neighbor information, a distributed control law is proposed for a group of unicycles to achieve the circular formation around the fixed target [151]. The cyclic pursuit strategy-based bearing measurements are applied to achieve the circular formation tracking for fixed target by a group of unicycles [152]. The localization and circumnavigation of the target with an unknown position are investigated in [153]. A similar problem is addressed in [55,122,154] where control algorithms are proposed based on the range and range rate measurements to circumnavigate an unknown target.

### 3.2.2. Moving Target Tracking

The proposed methodologies applied for circular formation enclosing for a fixed target cannot be directly extended to the case of a moving target [60]. The moving target enclosing the team of robots with relative position measurements is studied in [84] under the dynamically changing relative topology. In this problem, consider a group of $n$ agents modeled as (2) where each agent $i$ has two neighbors defined as $\mathcal{N}_i(z) = \{i^+, i^-\}$. The dynamics of the moving leader or target with unknown velocity are modeled as:

$$\dot{z}_0 = v_0, \tag{6}$$

where $z_0 \in \mathbf{R}^2$ and $v_0 \in \mathbf{R}^2$ represent, respectively, the target position and its piecewise constant velocity. To solve this problem, a local information adaptive controller is presented as follows [84]:

$$\begin{aligned} \dot{v}_i &= g_i((z_0 - z_i), (z_i - z_j)|_{j \in N_i(z)}), \\ u_i &= f_i((z_0 - z_i), (z_i - z_j)|_{j \in \mathcal{N}_i(z)}, v_i), \end{aligned} \tag{7}$$

where $v_i$ is the state of the feedback controller. The control law $u_i$ is designed based on the decoupling problem as follows:

$$u_i = u_i^1 + u_i^2, \tag{8}$$

where the term $u_i^1$ is designed for target following $\lim_{t \to \infty} ||z_i(t) - z_0(t)|| = r$ with $r$ the radius of the desired circular trajectory. The term $u_i^2$ is designed to maintain the desired distribution $\lim_{t \to \infty} ||z_i(t) - z_{i^+}(t)|| = \lim_{t \to \infty} ||z_j(t) - z_{j^+}(t)||$ for any $i, j$. In [155], a distributed cooperative control scheme based on cyclic pursuit method is introduced to achieve the uniform circular formation around moving target in 3D space. Moreover, the slowly moving target circular formation for agents modeled as simple integrators with unknown velocity is studied in [50]. Distributed control strategies are designed based only on bearing or distance measurements. Based on the cyclic pursuit technique, the circular formation tracking of a moving target for single and double integrators is investigated in [132]. By using bearing measurements, the localization and circumnavigation of a moving target by mobile agents is addressed in [156]. The desired circular formation around the target is achieved in the case that the velocity information of the target is known by one of the agents [156]. In [157], the enclosing control problem with respect to a moving target of unknown bounded velocity is considered. Further, the problem of enclosing a moving target with time varying velocity is studied under the assumption that the target's velocity is unknown to all agents [28].

For nonholonomic robots, the moving target tracking in CFP has been studied for a target with constant linear velocity in [63,116,158]; however, in this case, the target moves with constant velocity. The CFP becomes challenging for a moving target with time-varying velocity. A control algorithm based on translation control is proposed to stabilize the agent's trajectory in a circular motion with time-varying reference [159]; however, the proposed algorithm is less robust since it requires the initial conditions. In [160], a distributed control algorithm that distributes a group of unicycle-type vehicles

on even circular formation around a moving target with locally bounded tracking error is presented. With affine transformations and exosystem design, a translation control protocol is developed to achieve the circular formation stabilization around a moving target [48]. Based on the cyclic pursuit method, a cooperative control strategy using local measurements is proposed to achieve the evenly distributed circular formation tracking of moving target [161]. Based on backstepping techniques, two control schemes are designed to achieve the circular formation around a moving target with undirected sensor graph condition [93]. Finite-time circular formation control tracking of moving target by using the backstepping technique is investigated in [99] under a directed graph topology. In those approaches [48,93,99,161], the proposed algorithms require the target information such as (relative) position, speed, and acceleration. The same technique is used with a distributed control law to lead a group of nonholonomic mobile robots to rotate around a moving target with unknown velocity [162]. In this case, a tracking differentiator is applied to estimate the target velocity. The stable circular formation around the moving target is achieved in [59] with tracking errors within a small neighborhood of zero. Furthermore, similar to [48] circular formation tracking is studied in [37] with unknown target velocity. Circular formation control around a moving target is achieved in [60] where a control law is designed based on the position estimator instead of global position measurements. In [163], a multicircular circumnavigation around a moving target with time-varying velocity is studied under the assumption that at least one agent has the knowledge of the target information; however, the proposed controller in [163] cannot control the angular spacing among the agents and cannot guarantee that the tracking error converges to zero. To overcome these limitations, [41] proposed a consensus-based estimator control protocol such that the agents orbit a moving target in different radii. Recently, an adaptive estimator and control mechanisms are designed to guarantee the circumnavigation of a mobile target by unmanned surface vehicles (USV) [164]. Further, [165] proposed a bearing-based MASs distributed least square algorithm to circumnavigate a maneuvering target.

### 3.2.3. Tracking of Multiple Targets

In MASs, tracking multiple targets in achieving the formation is more significant and challenging. In the multiple targets tracking, since the center of the circular trajectory is not defined, the center estimation algorithm is required [166]. Unfortunately, it is not easy to estimate the center of the circular trajectory. A single agent with multiple targets using a circumnavigation-based technique is studied in [166–169] as shown in Figure 5a. In particular, an estimator for localization and control algorithms for system (2) are proposed for circular formation tacking problem for both stationary and slowly moving targets $z_{T_i}(t) \in \mathbf{R}^2$, $i = 1, \cdots, n$ by single agent with known trajectory $z_a(s) \in \mathbf{R}^2$ such that $s \leq t$ [166]. For estimator design, first the estimate of all target positions is needed $\hat{z}_{T_i}(t)$. The proposed estimator for stationary target $i$ is given as

$$\dot{\hat{z}}_{T_i}(t) = k_{est}(I - \varphi_i(t)\varphi_i^T(t))(z_a(t) - \hat{z}_{T_i}(t)), \tag{9}$$

and the control law as

$$\dot{z}_a(t) = u(t) = (\hat{r}(t) - r_d(t))\varphi(t) + \alpha\bar{\varphi}(t), \tag{10}$$

where $k_{est}$ and $\alpha$ are positive constants, $\hat{z}_T(t) = \frac{1}{n}\sum_i \hat{z}_{T_i}(t)$ represents the estimate of the center towards which the agent is seeking to move, $\hat{r}(t) = \|z_a(t) - \hat{z}(t)\|$, $\varphi(t) = \frac{\hat{z}_{T_i}(t) - z_a(t)}{\|\hat{z}_T(t) - z_a(t)\|}$ represents the bearing angle of $\hat{z}_T(t)$, $\bar{\varphi}$ represents the unit vector calculated by $\pi/2$ clockwise rotation of $\varphi$, and $r_d(t) = \max_i \|\hat{z}_T(t) - \hat{z}_{T_i}(t)\| + d$ with $d > 0$ represents the desired radius of the circle. It was shown that stability is guaranteed in stationary targets case. Further, the proposed algorithm allows for slow movements of the targets. Hence, the larger the speed of the moving targets, the larger the estimation error [166]. With bearing measurements, an estimator and control protocol is designed in [167] to

drive an agent to circumnavigate a group of stationary or moving targets. Based on the range measurements, sliding mode control algorithms are designed and justified to ensure the circumnavigation around multiple moving targets by the nonholonomic robot in 2D space [168] and 3D space [169]. Recently, a new circumnavigation pattern is developed by only using the bearing measurements to enable an agent to circumnavigate the convex hull of multiple targets [170].

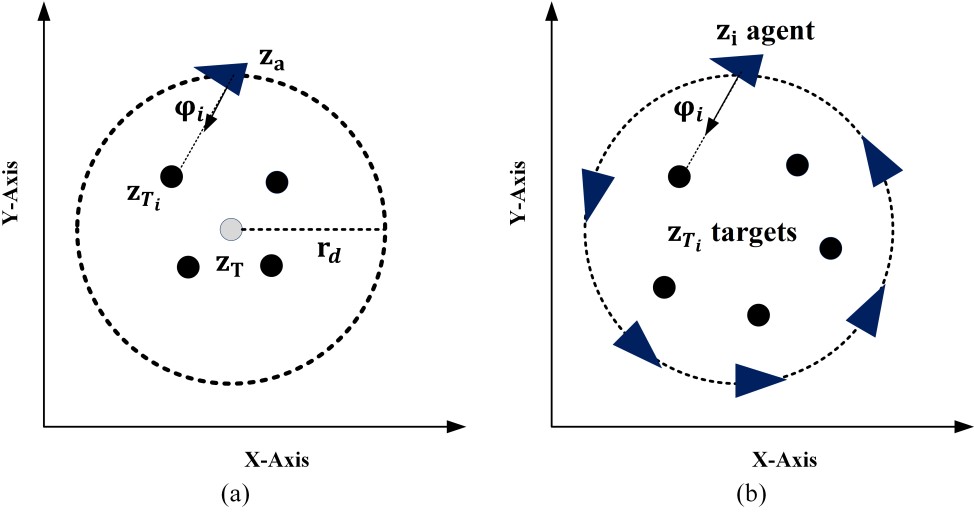

(a)  (b)

**Figure 5.** Circular formation: (**a**) a single agent with multiple targets; (**b**) group of agents with multiple targets.

In Figure 5b, a group of agents with multiple targets using a circumnavigation-based technique is investigated [29,58,171]. In this scheme, each target is enclosed by multiple robots. Compared with the single agent scheme, this scheme has several advantages such as flexibility, robustness, and diversity [99]. Based on the estimation technique, a novel distributed control strategy is proposed to achieve the uniform circumnavigation [171]. Agents move around multiple dynamic targets such that the graph communication topology contains a spanning tree [171]. The circular formation tracking of multiple leaders is also investigated for second-order MASs where the follower agents track the average state of multiple leaders with time-varying delays [172]. A target enclosing control protocol based on the center estimation algorithm and consensus is developed to achieve the cooperative equidistant circular formation by a group of agents [29]. Recently, in [58] the authors studied the localization and the circumnavigation of multiple targets with unknown positions by a group of agents. The design of the estimator and the control law for the above-mentioned scheme is presented in [56,166] as an extension to multiple targets and multi-agent cases. Furthermore, [173,174] considered the problem of a multi-target localization and formation circumnavigation on an elliptical trajectory by designing a geometric center estimator and novel circumnavigation control law. It assumes that the target's position and velocity are unknown for each agent. Note that the design of the circumnavigation controller and estimator for elliptical formation is complicated compared to circular formation.

### 3.3. Behavior-Based Circular Formation Control

In real-world applications, there exists many MASs such as satellite and UAVs where the agents may be required to perform several behaviors/tasks. The behavior-based methods have been introduced in [175] where the global task of MAS is divided into subtasks. This method is applied in case when the formation control requires many desired and expected behaviors from each agent such as the collision avoidance [98], obstacle avoidance [176], target seeking [59], formation keeping [94], localization [56], desired

collective centroid, and collective motion [33]; however, here the challenge is to design a control law for multiple tasks for each agent and achieve the global desired behavior.

A behavioral control scheme is designed in [81] to maintain the collective circular motion through following a moving beacon by a group of nonholonomic vehicles with sensory limitations. In [177], the authors study two collective behaviors the collective circular motion and the circumnavigation by a group of nonholonomic mobile robots. To accomplish this objective, a nearest-neighbor algorithm is developed based only on the range measurements to lead the agents to spread evenly on the prespecified circular trajectory. Moreover, in formation control the collision avoidance is fundamental to ensure that the agents do not collide in the formation and avoid the external obstacles [178]. A behavior-based approach is applied in [179] to achieve the circular formation with collision avoidance. Further, a behavior-based approach is addressed in [86] such that a group of autonomous MASs achieves the circular formation with collision avoidance. It is shown that the collision avoidance is guaranteed if the solution of system (2) satisfies $\|z_i - z_j\| > 0$ for any pair $i, j, i \neq j, t > 0$ and the order preservation is achieved. The research study [95] proposed a collision avoidance strategy with phase-coupled oscillators dynamics to achieve the symmetric circular formations without collision. This problem is studied in networking techniques, such as ring topology, etc. The proposed control law depends on the local measurements in the Frenet–Serret frame for every vehicle. In [115], the HDTA algorithm was designed to solve the problem of encirclement control of multiple robots and multiple targets. The problem combines three interlinked behaviors of multi-robot: (a) the task allocation of a group of moving targets, (b) encirclement control of a group of agents around a target, and (c) collision avoidance. A collision avoidance methodology is proposed in [57] to achieve the circular formation tracking of a moving target in 3D space. Based on sliding mode control, a collision-free navigation strategy is proposed for mobile robots to solve the problems of border patrolling missions [180]. The authors in [181] proposed a guidance control law by using sliding mode control for a unicycle-like vehicle to achieve border patrolling while assuring obstacle avoidance.

### 3.4. Circular Formation Using the Artificial Potential Functions

The artificial potential has been used widely in the stabilization of the formation control problems in MASs. This method has been proposed first in [182] for the obstacle avoidance of multi-mobile robots. This approach constitutes a negative gradient containing both the attractive and repulsive potentials as input to guarantee the global convergence and collision-free formation [183]. The artificial methods are also used to achieve the uniform and even distribution, rotation around target on the circular formation. The critical points of the potential functions in circular formation are established in [117].

A vector field-based artificial potential has been applied in circumnavigation problems for the mobile robots in [64,116,118] where the vector field globally leads the agents to a limit cycle around the target. The vector field can be created by the addition of the circulation to the gradient field of the potential function. An extension using orthogonal vector field for circumnavigation problem for a moving target in 3D is presented in [184]. This is further extended to [98] where collision avoidance is considered. Based on rigidity graphs, the LGVF approach is proposed to lead a group of unmanned aircraft to move on spaced circular formation around moving target [63]. In this approach, the vector field was assumed stationary and centered at the origin. The general techniques for the construction of Lyapunov vector field circular formation attractors are presented in [62]. Based on the vector fields approach, a distributed estimation control law is developed for a multi-robot system to enclose a moving target with time-varying velocity in evenly distributed circular formation for patrolling mission [185]. This control law does not require access to the full target information.

A special type of artificial potential called the navigation function has been applied in CFP. This technique avoids the local minima by integrating the negated gradient vector field of the cost function [186]. This leads to trajectories ensuring the collision avoidance [186].

Its control design is applied for collision avoidance and external obstacles. To address realistic communication objectives a modified navigation function with multiple minima was presented in [187]. The conditions for circular formation target tracking using the same technique are relaxed in [188] with the following decentralized navigation function:

$$\varphi_i = \frac{\alpha_i}{(\alpha_i^k + \beta_i)^{\frac{1}{k}}},$$

(11)

where $k \in \mathbf{R}^+$, $\alpha : \mathbf{R}^2 \to \mathbf{R}^+$, $\beta_i : \mathbf{R}^2 \to [0,1]$ represent the tuning parameter, the goal function, and the obstacle avoidance function. This guarantees the uniform spacing positions of the agents in the target tracking problem for multiple UAVs. A combination of attraction/repulsion potential functions are applied to achieve a general uniform circum-navigation around the target in common orbit [189] and different orbits [21]. By using BLF with bounded phase potentials, control protocols are designed to stabilize phase coordinated patterns of unicycle-type around a prespecified common circular trajectory while keeping their trajectories within a prespecified circular boundary [190]. A novel Lyapunov non-uniform approach based on vector fields is proposed in [191] that enables an entire group of aircraft to follow a circular path around the target while making a formation with relative phase-shift angles between UAVs. Recently, a novel guidance control law is designed for multiple UAVs to circumnavigate a ground-moving target [192]. The authors in [193] proposed a robust control technique by using the backstepping approach to solve the circumnavigation problem of group heterogeneous MASs.

### 3.5. Virtual-Structure-Based Circular Formation Control

The virtual structure method has been used widely in formation control of MASs [194]. The notion of the virtual structure was studied first in [195] where a general control scheme is constructed to force the agents to form a single rigid structure with no hierarchy. The objective of this method is to design control algorithms to minimize the error between the real position and the virtual position of the agents. An illustration of the virtual structure circular formation is represented in Figure 6. This method is applied to achieve the circular formation such that all the agents form a rigid structure based on a virtual reference point or virtual leader [97].

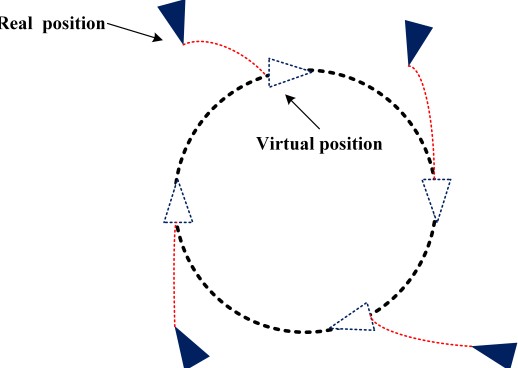

**Figure 6.** An illustration of virtual-structure-based circular formation method.

A distributed coordination control scheme based on the virtual structure to achieve multi-robot circular formation is presented in [196]. A virtual structure introduced in [119] solves the target tracking problem for multiple nonholonomic agents. It was shown that the proposed strategies ensure the circular formation tracking of the moving target. By using consensus-seeking strategy, a virtual structure circular formation is achieved while tracking a moving target with dynamic graph topology [120]. A virtual structure approach is applied in [97] where a virtual robot is considered as the center of the circular trajectory such that the mobile robots move into the circle. To achieve stable navigation in a multi-robot system depending on constrained virtual structure a circular formation control scheme is proposed

in [61]. In [96], a distributed control algorithm based on the virtual structure approach is designed for multiple fixed-wing UAVs to achieve circular formation. To realize this goal a virtual UAV is maintained as the center of the formation. The virtual structure method for UAV circular formation control is discussed in [197].

## 4. Collective Circular Motion Stabilization

This section reviews the stabilization problem for circular formation where the center is not specified and it is dependent on the initial states of agents [17,18,22,198]. Based on literature, different control techniques are proposed to achieve CCM in MASs [199,200]. Some CCM control strategies aim to stabilize the agents with the same radius/circular path while the rest discuss the stability and formation issues of agents with different radii/circular paths. The orientation of such CCM strategies is presented in Figure 7a,b. For nonholonomic vehicles, many researchers studied the circular formation stabilization problem using the kinematic unicycle model [23]. The mathematical model for the kinematic unicycle model is presented as [17,22,23]:

$$\dot{x}_i = v_i \cos(\theta_i), \ \ \dot{y}_i = v_i \sin(\theta_i), \ \ \dot{\theta}_i = w_i, \ \ i = 1, \dots, n, \tag{12}$$

where $z_i = [x_i, y_i]^T \in \mathbf{R}^2$ and $\theta_i \in \mathbf{R}$ represent the position and the heading angle of agent $i$, $v_i \in \mathbf{R}$ and $w_i \in \mathbf{R}$ are, respectively, the linear and the angular velocities representing the control inputs for each agent $i$. Which is equivalent to

$$\dot{r}_i = v_i e^{i\theta_i}, \ \ \dot{\theta}_i = \omega_i, \ \ i = 1, \dots, n, \tag{13}$$

with $r_i = x_i + iy_i = |r_i|e^{i\phi_i} \in \mathbf{C}$, $\phi_i = \arg(r_i)$. For $i = 1, \dots, n, r_i, \theta_i$, and $v_i$ are, respectively, the position, orientation, and velocity of the agent $i$.

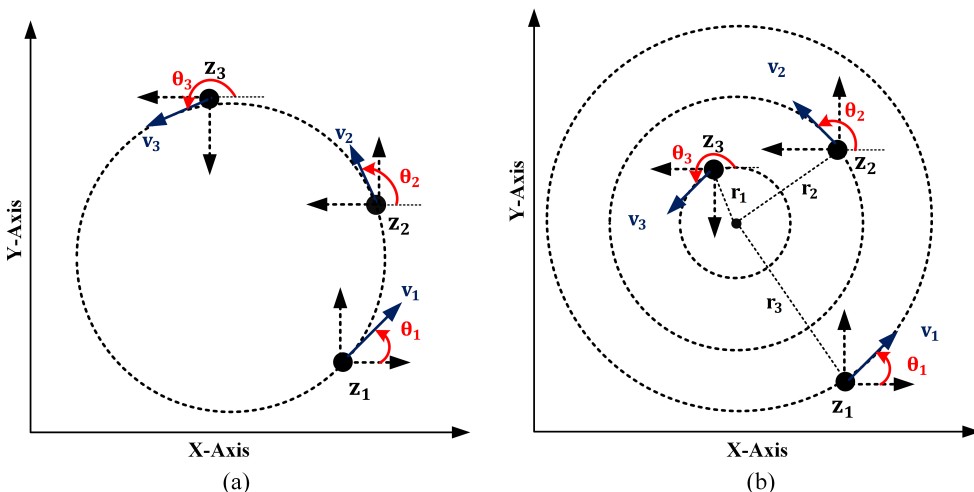

**Figure 7.** Collective circular motion patterns: (**a**) collective circular motion in the same radius; (**b**) collective circular motion in different radii.

### 4.1. CCM with Identical Linear Velocity

The extension of cyclic pursuit strategy [111] is used to develop a kinematic unicycle model [17] to achieve CCM. The stabilization to achieve CCM is performed by applying the linearization technique to those strategies. It is shown that the equilibrium in formations control problem is achieved by network communication to generalized regular polygons. Control algorithms in [110,201] discussed that the speed of each unicycle is proportional to the distance from the next unicycle. Hierarchical control algorithm for the agents with dynamic unicycle model is presented in [18]. The proposed scheme achieves stabilization in CCM. A necessary condition for the proposed control scheme requires that the graph topology of the unicycles contains a spanning tree. A MASs consensus technique using

distributed bearing-only control law is designed to achieve the balanced circular formation with collision avoidance [44,45]. A cyclic pursuit control law based only on the relative position of agent to achieve CCM is proposed in [34]. It was proven that all the trajectories of unicycles are bounded by using the pseudo-linearization technique; however, a leader-free distributed algorithm is designed to achieve the CCM for a group of nonholonomic agents [22]. The system has a limited communication range and a common rotational radius. The proposed control protocol takes the following form [22]:

$$
\begin{aligned}
v_i =& r\omega_i - \sum_{j \in \mathbf{N}, j \neq i} \mu(||z_{ij}||)[\cos\theta_i \ \sin\theta_i] \begin{bmatrix} x_{ij} - r\xi_{ij} \\ y_{ij} - r\tau_{ij} \end{bmatrix} \\
\omega_i =& \omega_0 + \sum_{j \in \mathbf{N}, j \neq i} \nu(||z_{ij}||)sgn^+(\sin\theta_{ij}), \ i \in \mathbf{N},
\end{aligned}
\tag{14}
$$

where $r := v_0/\omega_0$, $\omega_0$ and $v_0$ represent the rotational and forward velocities of each agent, $\tau_{ij} := \cos(\theta_i) - \cos(\theta_j)$, $\xi_{ij} := \sin(\theta_i) - \sin(\theta_j)$, $z_{ij} := z_i - z_j$, $x_{ij} := x_i - x_j$, $y_{ij} := y_i - y_j$, $\theta_{ij} = \theta_i - \theta_j$. The terms $\mu$ and $\nu$ represent the interactions among every pair of neighboring agents. The functions $\mu(\cdot), \nu(\cdot)$ belong to class $\mathcal{S}$ functions. The designed control algorithm ensured the collective circular formation when the joint connectivity condition is satisfied. An extended distributed algorithm is designed for a group of heterogeneous agents to move on different circular trajectories around a common center [198]. A backstepping control algorithm is designed for the stabilization of circular formations of self-propelled vehicles with second-order rotational dynamics [202]. A gradient control using artificial potential based on balanced graph condition is designed in [19] to achieve CCM with identical constant angular velocities. Based on relative position measurements, an observer-based feedback control protocol is designed for the stabilization of the CCM with desired angular velocity in [203].

**Remark 2.** *Note that the stabilization of the synchronized and balanced formations require the average linear momentum of the group of N agents called the phase order parameter $P_\theta$ [23,42,43] defined as:*

$$
P_\theta = \frac{1}{N} \sum_{k=1}^{N} e^{i\theta_k}.
\tag{15}
$$

*The minimization of phase order parameter leads to a synchronized formation (i.e., $|P_\theta| = 1$ ) and its maximization leads to a balanced formation (i.e., $P_\theta = 0$ ).*

**Remark 3.** *There is a relation between the radius of the circular trajectory r and the linear velocities $v_i$ of the agents for $i = 1, \ldots, n$ given by*

$$
v_i = r\omega_0.
\tag{16}
$$

*Hence, all the agents move on the same circular trajectory of radius $r = v_0/\omega_0$ if and only if they achieve a common linear velocity $v_0$.*

### 4.2. CCM with Nonidentical Linear Velocity

The control schemes with the assumption of identical velocities in unicycle models do not satisfy practical application [43]. With nonidentical velocities, it is challenging to achieve the CCM of unicycles on a common circular trajectory with constant angular frequency [23]. Stabilization of planar CCM is proposed in [33]. An extension to CCM stabilization with reference to motion coordination problem for a group of heterogeneous unicycles (12) with nonidentical constant forward speeds is solved under the fixed gain

control input in [23]. To stabilize the CCMs into a common circular trajectory, a control law is presented as follows [23]:

$$u_i = \omega_i(1 + \alpha < \tilde{r}_i, \dot{r}_i >) + \alpha v_i^2 \frac{\partial U_{P_\theta}}{\partial \theta_i}, \tag{17}$$

where $\alpha > 0$, $r_0 > 0$ is the radius of the desired circle, $v_i > 0$, $i = 1, \ldots, n$ represents the nonidentical velocities, $\omega_i = v_i/r_0$ denotes the angular frequencies, $\tilde{r}_i = r_i - R$, $R = 1/n \sum_{i=1}^{n} r_i$, $\frac{\partial U}{\partial \theta_i} = < ie^{i\theta_i}, P_\theta >$ with $P_\theta$ defined as (15). This problem has been studied with complete graph condition under two different frameworks, the CCM with different radius and common angular frequency and the CCM with common radius and different angular frequency. This problem is studied under restricted communication modeled as time invariant such that the agents move on a common circular trajectory or different circular trajectories with prespecified angular velocity. Furthermore, a centralized circular formation algorithm is proposed in [43] to revisit the problem proposed in [23] with a variable gain control approach. The numerical optimization approaches BFGS method [204] and LM method [205] are utilized based on variable gain control. The developed technique is beneficial in reducing the workload in the system. The discussed methods are applied to solve the balanced circular motion formation and the concentric circular formation respectively. It is worth noting that most of the proposed control strategies are based on the fixed gain control input to achieve the stabilization of the unicycles [43].

## 5. Event-Triggered Circle Formation Control

Event-triggered control (ETC) of MASs is studied in [206]. The ETC strategy is divided into centralized case and decentralized case. Under the centralized ETC, the global error is adopted to update the control input and each agent has its event-triggered condition. This gives information on when to update the controller for a particular operation. In the decentralized ETC, the control input is updated based on the agents and their relative information; however, this strategy is proposed in the case where the control updates are determined by a series of certain events. In ETC, we say that an event is triggered when the magnitude of a certain measurement error achieves the specified threshold. The key feature of applying the ETC is that the communication sources and costs are saved effectively since it decreases the energy consumption of the systems by reducing the control updates.

This control scheme has been applied recently to the circular formation control problem where the ETC problem is transformed as agreement problem [207]. Generally, an ETC-based CFP can be stated simply as follows: A circular formation of MAS determined by $p = [p_1, p_2, \ldots, p_n]^T$ such that $p_i > 0$ and $\sum_{i=1}^{n} p_i = 2\pi$. The design of control law $u_i(t), i = 1, \ldots, n$ is developed by using an event-triggered scheme for each agent $i$ using only its own and its two neighbors' states $i^+$ and $i^-$. Each agent state in ETC transmits according to a certain triggering condition. The initial conditions for such a scheme are: $0 \leq z_1(0) < \ldots < z_n(0) < 2\pi$. The solution of the system (2) (i.e., $\dot{z}_i(t) = u_i(t)$) asymptotically converges to some equilibrium point $z^*$ satisfying $y^* = p$. Under this framework, the distributed ETC circular formation problem has been studied for system (2) in [207]. Control algorithm for asynchronous event-triggered communication is designed to achieve circular formation. Assume that the time sequence for each agent $i$ is denoted by $t_0^i, t_1^i, \ldots$. Then, the measurement error [207] can be described as:

$$e_i = y_i(t_i^k) - y_i(t), \ t \in [t_k^i, t_{k+1}^i) \tag{18}$$

and the corresponding controller is:

$$u_i = \sum_{j \in \mathcal{N}_i^{out}} \frac{p_i}{p_i + p_j} (y_j(t_{k'_j(t)}^j) - y_i(t_k^i)) \tag{19}$$

where $y_i(t) = \begin{cases} z_{i^+}(t) - z_i(t), & i = 1, \ldots, n-1; \\ z_{i^+}(t) - z_i(t) + 2\pi, & i = n, \end{cases}$ and $k'(t) = \arg\min_{l \in \mathbf{N}: t \geq t_l} \{t - t_l^j\}$. It is shown that under the proposed control framework the Zeno behavior is excluded. Note that the centralized ETC has been applied first to the first order MAS to achieve circular formation in [208]. Moreover, the problem is extended in [73] to second-order CFP. This extension made the design more realistic and applicable by adding the velocity variable and applying centralized and decentralized event-triggered strategies, respectively [73]. It is shown that the decentralized case does not guarantee the strictly positive condition of inter-event intervals [73]. The above-proposed ETC schemes are studied under the continuous-time framework. A distributed ETC circular formation of MASs is investigated under the discrete-time framework in [74]. In this technique, the communication bandwidth has been taken into consideration. In this framework, a distributed control law is proposed based on the combination of the ETC mechanism and the quantized control technology. The bearing rigidity and ETC mechanism are applied in [209] to address the encirclement control problem of MAS around a moving target. Further, [210] addressed the ETC-based CFP for a group of anonymous mobile robots with restricted computational and communication abilities. Recently, a novel co-design combination of nonuniform quantization and ETC mechanism is designed in [211] to solve the ETC-based CFP. In [212], ETC mechanisms are proposed for an under-actuated ASV to circumnavigate a moving target whose velocity is unknown. Further, [213] proposed a decentralized ETC strategy based on periodic sampled data to solve the decentralized ETC-based CFP with or without time delays.

## 6. Circular Formation Control with Constraints

In real-world applications, many devices are subject to different constraints such as input/output constraints [214]. In order to address real applications and increase the applicability of circular formation, the CFP has been studied under different constraints such as actuator saturation [68], actuator fault [215], disturbances[70], velocity constraint [20], finite-time control [39], etc.

### 6.1. CFP with Actuator Saturation

A distributed control law under identical input saturation constraints is proposed in [69]. The control law drives a group of first-order agents with discrete-time dynamics to achieve circular formation. The distance is considered consistent between the neighboring agents in this scheme. It is observed that the spatial order preservation of agents is independent of input saturation constraints. A coverage control method has been applied on a circle for a class of first-order heterogeneous mobile sensors with input saturation [68]. This method drives the sensors to optimal configuration such as the preservation of the spatial order and the collision avoidance on the circle is ensured [68]. A distributed coverage control strategy has been applied to solve the circular formation with input saturation for group of agents with bounded measurement errors [216]. The main idea behind using the coverage control comprises of minimizing the coverage cost function, which is considered as the largest arrival time from the mobile sensors to the points located on the circle. Moreover, a distributed control scheme has been proposed to achieve the semi-global leaderless consensus for a group of agents moving in a circular trajectory subject to input saturation [217]. The proposed scheme is designed based on the low gain feedback technique and studied under switching topology. The circular formation is studied with different input saturation constraints and limited interaction in [218]. Recently, based on the Fisher information and the neural networks, [219] proposed an optimization criterion and an optimal circumnavigation algorithm to solve the problem of the optimal circumnavigation of a UAV around a mobile target in the presence of input saturation.

### 6.2. CFP with Actuator Fault

In formation control of MASs, the possibility of suffering from communication topology faults increases with system complexity [220]. The CFP with actuator failures is

investigated in [215]. An adaptive control mechanism is proposed to achieve the time-varying circular formation tracking of agents. The system is modeled as second integrator MASs under the actuator failures and non-cooperative target. It is shown that the follower agents achieve the desired rotating circular formation while following the trajectory of the leader.

### 6.3. CFP with Disturbances

The circular formation of mobile agents subject to constant disturbances is studied in [70]. A distributed control law is designed to achieve the desired circular formation for agents initially distributed on the circle. Later, the CFP with heterogeneous input disturbances is studied in [221]. A dynamic control law using relative displacement measurements is proposed. The agents are modeled as a kinematic unicycle model to achieve the evenly spaced circular formation around a fixed target.

### 6.4. CFP with Communication Constraint

In this scheme, the agents have limited sense. Agents are required to interact within a certain communication range. This issue has been solved in [222,223]. In this technique, CFP is considered under a limited interaction range in a counterclockwise direction. A distributed switching control law is designed for such CFP [222]. It is proved that the desired circular formation is achieved if the agent interaction range parameter is large enough than the angular distance between the agents. In [223], a distributed coverage controller is designed by using low gain feedback technique. CFP under bandwidth limitation of communication is studied in [224] using an encoder–decoder control scheme.

### 6.5. CFP with Locomotion Constraint

To achieve the circular formation subject to locomotion constraint, a distributed control protocol based on sampled-data control strategy is designed in [40]. The problem is focused on the time delay case. In this case, the agents can only sense the relative angular positions. The necessary and sufficient conditions are derived to ensure the asymptotic stability of the system. The agents are supposed to only move forward but not backward in a counterclockwise direction on the circle. Such a problem describes practical applications such as the Dubin's vehicles [225], biomimetic robots [226], etc.

### 6.6. CFP with Velocity Constraint

In practical applications, MASs suffer from velocity constraints caused by stall conditions and thrust limitations e.g., fixed-wing (UAV) [227]. Hence, it is a practical requirement to study the formation control under velocity constraints. CFP with velocity constraint is studied in [20,80,228,229].

### 6.7. CFP with a Finite Time Control

Practically, the circular formation requires to be achieved in finite time. The finite-time control has many advantages compared with infinite-time control. The advantages include disturbance rejection, faster convergence, and robustness [230]; however, only a little research has been reported on the finite-time CFP. For instance, the design of finite-time control algorithms for CFP with single integrator MASs is addressed in [39,229,231]. In [39], the finite time CFP is solved based on the time-varying gain technique. A cascade control technique is presented in [229] to solve the circular formation tracking of moving targets with directed topology and velocity constraints. Authors in [231] achieved the circular formation tracking of a group of stationary targets using the estimation technique. Moreover, the finite time CFP for nonholonomic robots is studied in [99] where finite time tracking and agreement controllers are developed separately. Recently, various novel control schemes have been designed to ensure the circumnavigation within finite time [232,233].

It is worth mentioning that only a little research work has considered the CFP with constraints and most existing results are studied only for holonomic robots initially located on the circular trajectory.

## 7. Source Seeking via Circular Formation

The objective of the source-seeking problem is to detect and follow the source of some signals by a group of agents. The applications include pollution sensing [234], odor source detection [235], etc. Most of the proposed control protocols use the gradient-descent methods to deal with the source localization [236]. In this case, the gradient descent algorithm is developed by the gradient of signal strength. Whereas in most of the applications the gradient information is unknown [237]. In this case, the gradient can be estimated by two different techniques of spatially distributed measurements. A single-vehicle with changing position over time can take all the measurements [238]. The group of vehicles cooperates to take the measurements at different locations [16]. The uniform circular formation guarantees the estimation of the gradient at the center of the circular trajectory. For this reason, it is necessary to stabilize the agents into a circular formation. Based on the present literature [239], the gradient of the signal strength is estimated by the circular formation by using the simple weighted average of the measured signal.

Different source-seeking algorithms via the circular formation have been addressed in [71,237,240]. In [240], a source seeking algorithm based on an outer control loop is designed to achieve the following control objectives:

1.　Stabilization of the circular formation;
2.　Estimation of the gradient;
3.　Steering the circular formation with dynamic center toward the source for $n$ agents.

The agents are modeled as a kinematic unicycle model where the gradient direction is approximated by direct measurement of the signal strength. The drawback of this method is the assumption of the spatial propagation of the signal [237]. Moreover, a distributed algorithm based on the multidimensional Newton–Raphson consensus technique is used to solve the source localization problem for a group of vehicles. The vehicles are uniformly distributed in circular trajectory [71]. The suggested technique steers the center of the circular motion of agents by using the estimated gradient of the measured signal [71].

## 8. Applications of Circular Formation

The circular formation has a variety of practical applications in industry. The circular formation of autonomous satellites is an active and challenging research task. The task involves the movements of the autonomous satellites in the orbit at given speeds while avoiding collisions. The communication of those systems to share information is also a challenging task. In circular formation, each orbiting satellite communicates with a limited set of neighbors (limited connectivity) [241]. It has application in the defense industry to provide surveillance and navigation of a particular field with defined radius [242]. Applications in escorting and patrolling missions of multi-robots for example UAVs patrolling geographic borders [243,244]. The control design and stability of UAVs and UGVs to follow a circular trajectory is an interesting research paradigm. The control challenges are obstacle avoidance, collision avoidance, and trajectory/target formation [245,246].

Practical applications of CFP such as PID and pole placement control law stabilize a micro-satellite formation flying in a circular earth orbit [247]. Circular formation-based micro-satellite navigation and surveillance are presented in Figure 8a. The formation keeping of satellites in a circular orbit is studied based on sampled-data representations of satellites [248]. Different satellite formation flying missions have been realized among which are the CanX-4 and CanX-5 missions forming a projected circular orbit [249]. Currently, a CCN control design is proposed for the cooperative circumnavigation of multiple micro-satellites around a host spacecraft [250].

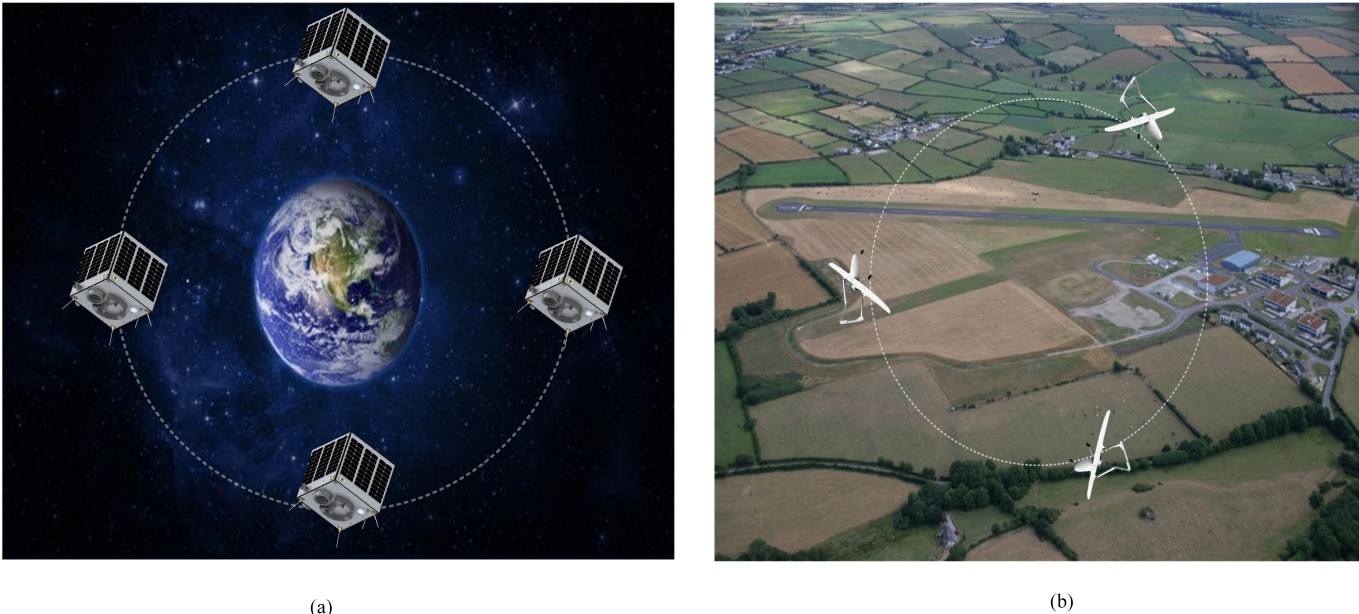

(a)

(b)

**Figure 8.** Circular formation applications: (**a**) group of micro-satellites orbiting around Earth; (**b**) group of UAVs surveillance and navigation mission.

In the Earth-observing missions, the UAVs are required to perform a circular formation flight around the Earth in special orbit [251]. Moreover, a circular formation flight is achieved for a cooperating team of UAVs orbiting around a ground target subject to the wind with unknown velocity [252]. A circular formation flight control is designed for UAVs around a set of circles in the presence of spatiotemporal disturbance and directed networks [253]. A leader–follower control method is applied in [254] for solving the problem of circular formation flight for multiple quad-rotor aerial vehicles (QAV). Currently, a CCN control problem has been studied for groups of networked UAVs, in which the circumnavigation is achieved on multiple spatial orbits with different radii [255]. In Figure 8b UAV circular formation based surveillance is presented.

The circular formation of car-like robots with limited sensing is realized while enclosing a static and moving target in [256]. The experimental platform and the practical implementation of CCM on MASs composed of three Pioneer 2-DXe bi-wheeled robots are presented in [257]. The circular formation control also has a potential interest in circumnavigation. An LQT controller is designed for linear and non-linear models to achieve the fast circumnavigation missions [258]. Applications of circular formation in complex missions are surveillance missions [259], the source seeking missions [24], and cooperative localization [104]. Cooperative circumnavigation of quadrotor equipped with the onboard downward-looking camera around an unknown target is studied in [260]. Future trends in the design and analysis of orbiting vehicles involve the concepts of circular formation control design and stability analysis.

## 9. Future Research Directions

In the past few years, the circular formation problem of MASs has been widely studied by different research communities. Since the study of the circular formation problem has more practical significance, many control schemes have been developed to meet the practical requirement of the circular formation problem; however, there exists more challenges since the existing research control schemes have some limitations in practical applications. Many issues and problems remain open in the circular formation control of MAS. Discussions on further issues can be summarized as follows.

- Many results in CFP consider the single-integrator model; therefore, it is a practical requirement to address more complex and realistic MAS models such as models that capture real UAV systems with dynamic uncertainties and noisy environments.
- Although the existence of several extensive research papers, CFP investigations mainly focus on identical (Homogenous) MAS; however, in some practical applications such as reconnaissance missions carried out by UAVs and UGVs, it is difficult to consider that all the agents have the same dynamics. Thus, more research effort on circular formation control of heterogeneous MASs will be needed.
- For real-time circular formation, practical issues, such as connectivity preservation, obstacle avoidance, and collision avoidance, have been less studied in the circular formation of MAS. How to implement the rotational control input and design novel control strategies to ensure collision avoidance and obstacle avoidance among agents is a challenging research topic.
- The design of novel control strategies to address more practical and realistic scenarios such as CFP of MAS with switching communication topologies, unreliable links, and time-varying delays needs further investigation. Moreover, most of the existing research results on circular formation have focused on MAS in the plane, and the study cannot be directly extended to 3D space. Thus, the CFP of MAS in 3D space needs to be studied.
- To the best of the author's knowledge, there are only a few results in CFP with constraints, and most existing results consider only holonomic robots initially located in the circular trajectory; however, it is necessary to study the case where the agents are randomly deployed in the plane.
- Instead of wireless communication, the design of control law with only onboard sensors for example image processing is an open future research paradigm. Furthermore, the implementation of existing results of the CFP on a real multi-robot test to verify the theoretical achieved results needs to be addressed.
- In some practical formation flights, it is a requirement to achieve circumnavigation and localization in different circular orbits around targets; however, the existing results on the CFP of MASs have focused on circumnavigating targets in the same radii/orbit. How to design novel control strategies to achieve the circular formation tracking of targets in different circular radii/orbits is an open research problem that needs further investigation.

## 10. Conclusions

A brief review of recent advances in the circular formation of MASs is presented in this survey. A review of basic circular formation, its classification, and applications are presented. Leader–follower, virtual structure, cyclic pursuit, collective circular motion, and circumnavigation are the most studied techniques in this context. Many research directions such as realistic agent modeling and controller design, real-time collision avoidance, communication faults, and delays in multi-agent formation control present a wide gap in the research.

**Author Contributions:** Conceptualization, H.L.; methodology, H.L. and A.H.; writing—original draft preparation, H.L.; writing—review and editing, A.H. and Z.-Y.H. All authors have read and agreed to the published version of the manuscript.

**Funding:** This research received no external funding.

**Conflicts of Interest:** The authors declare no conflict of interest.

**Abbreviations**

The following abbreviations are used in this manuscript:

| | |
|---|---|
| CFP | Circular Formation Problem |
| CCM | Collective Circular Motion |
| MAS | Multi Agent System |
| CCFP | Centralized Circular Formation Problem |
| DCFP | Decentralized Circular Formation Problem |
| RSS | Received Signal Strength |
| HDTA | Hybrid Dynamic Task Allocation |
| LGVF | Lyapunov Guidance Vector Field |
| BLF | Barrier Lyapunov Function |
| BFGS | Broyden Fletcher Goldfarb Shanno |
| LM | Levenberg–Marquardt |
| ETC | Event Triggered Control |
| CCN | Cooperative Circumnavigation |
| LQT | Linear Quadratic Tracking |
| UAV | Unmanned Aerial Vehicle |
| UGV | Unmanned Ground Vehicle |
| AUV | Autonomous Underwater Vehicle |
| ASV | Autonomous Surface Vehicle |

**Appendix A**

The control law (5) is simplified as follows:

The first part $u_i^p(t)$ is designed as a limit-cycle oscillator as

$$u_i^p(t) = \lambda \begin{bmatrix} \gamma l_i(t) & -1 \\ 1 & \gamma l_i(t) \end{bmatrix} (z_i(t) - z_0(t)), \quad i = 1, \dots, n, \tag{A1}$$

for $\lambda, \gamma$ positive constants, $l_i(t) = r^2 - || z_i(t) - z_0(t) ||^2$ where $r$ is the desired radius. The second part $h_i(t)$ is designed as

$$h_i(t) = k_1 + \frac{k_2}{2\pi} f_i^\alpha(t), \quad i = 1, \dots, n, \tag{A2}$$

where $k_1 \geq k_2 > 0$, $f_i^\alpha(t) = \frac{d_{i^-}}{d_i + d_{i^-}} \widehat{\alpha}_i(t) + \frac{d_i}{d_i + d_{i^-}} \widehat{\alpha}_{i^-}(t)$ such that

$$i^- = \begin{cases} n & when\ i = 1 \\ i - 1 & when\ i = 2, 3, \dots, n \end{cases}, \quad i^+ = \begin{cases} i + 1 & when\ i = 1, \dots, n-1 \\ 1 & when\ i = n \end{cases}, \tag{A3}$$

$\widehat{\alpha}_i$ denotes the angular distance from agent $i$ to $i^+$, $\widehat{\alpha}_{i^-}$ denotes the angular distance from agent $i^-$ to $i$, $d_i$ is the desired angular spacing from agent $i$ to $i^+$, and $d_i^-$ is the desired angular spacing from agent $i^-$ to $i$.

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
