# Peer review of "A Survey on Techniques in the Circular Formation of Multi-Agent Systems"

_electronics, doi:10.3390/electronics10232959_

Round 1

Reviewer 1 Report

This paper performs a literature review of circular formation control. This is indeed a rich area where a lot has been done and thus review papers are indeed useful for people entering this area of research or just willing to keep up with the most recent results. Hence, I think this work can prove useful to a wide readership once the following issues have been addressed:

  1. The introduction spends too much time in summarizing the content of the paper, while the motivation to this review is only marginally explained. What gap does this review fill with respect to the existing ones? Does it just cover the most recents results or are there other differences?

  1. The authors cover most of the theoretical and applied branches of the line of research on circular formation control. However, in my view, a few more references and a little bit more of attention could be devoted to the patrolling and surveillance applications (only one reference on patrolling is included).

    From a theoretical perspective, the authors have missed the work on cyclic pursuit in the presence of bounded measurement noise and proximity sensors, see e.g., 

    Iudice, F. L., Acosta, J. Á., Garofalo, F., & DeLellis, P. (2020). Estimation and control of oscillators through short-range noisy proximity measurements. Automatica, 113, 108752.

    Interestingly, this framework can be used to adapt the control laws used for the circle, to the cyclic pursuit on a closed curve that is not the circle, see

    Iudice, F. L., Garofalo, F., & De Lellis, P. (2021). Formation control on a closed curve under an intermittent measurement flow. IEEE Control Systems Letters.
  2. This brings me to another key point. A single sentence is devoted to possible future work in this area. I expect more given from a literature review. A possible direction indeed could be adapting existing laws to the case where the agents don’t move on a circle but rather another closed curve. I expect the authors to have many more in mind.
  3. Overall the paper is readable. However, there are a large number of typos. Some are listed in what follows.

Line 55 measurement disorders, should this be measurement noise? 

Line 119 (1) is not a problem but a model. The authors themselves say before (1) that they are about to formulate a problem setup, i.e. a general model for the dynamics of the MAS that must achieve a circular formation.

Line 129 the authors must mean high reliability not low as they correctly say in line 130 that centralized solutions are not reliable and thus decentralized ones are.

Line 142-143 have been addressed should be have addressed.

Line 148/149 What is the ‘circumference of circle?)

Line 147: should even distribution be balanced formation?

Line 198 Different than should be Different from

Line 310 ‘functions, a distributed’ should be  ‘functions, distributed’ (the a should be omitted)

Line 315 the should be no period after data.

Line 324 algorithms should be algorithm

Line 330 contain should be contains

Line 333 ‘subject at’ should be ‘subject to a’

Line 335 ‘for class’ should be ‘for a class’

Line 337 ‘control schemes that steered’ should be ‘control scheme that steers’

Line 346 the line numbering stops for half a page

Line 354 ‘only on the bearing’ should be ‘only on bearing’

Line 369 ‘algorithms’ should be ‘algorithm’

Line 392 ‘this limitations’ should be ¡these limitations’

Line 395 again the numbering stops

Line 397 ‘the center where agent seeking to move’ should be ‘the center towards which the agent is seeking to move’  

Line 401 ‘allows a’ should be ‘allows for’

Line 402 ‘the larger speed of the moving targets imply the larger estimation’ should be ‘the larger the speed of the moving targets, the larger the estimation’

Line 403 ‘are designed’ should be ‘is designed’

Line 414 ‘contains spanning’ should be ‘contains a spanning’

Line 425 ‘may require’ should be ‘may be required’

Line 440 ‘to’ should be ‘do’

Line 450 has the acronym HDTA been defined?

Line 483 is the word ‘approach’ missing? A novel Lyapunov non-uniform what?

Line 509 again the line numbering stops In the fifth line of the first paragraph of section 4 the first ‘as’ should be omitted.

Remark 2: the word ‘the’ is missing before the word ‘group’

Line 527 again the word ‘the’ is missing before the word ‘sam

Line 546 the ‘Advance’ should be ‘Advanced’

Line 558 the numbering stops. In the following line ‘is applied’ should be ‘has been applied’

Line 604 ‘is increasing should be ‘increases’

Line 609: ‘of leader’ should be ‘of the leader’.

Line 637 ‘is practical’ should be ‘is a practical’

Line 651 ‘noval’ should be ‘novel’

Line 686 ‘has variety’ should be ‘has a variety’

Line 691 ‘with limited’ should be ‘with a limited’

  1. In general phrases as ‘it is hard and difficult’ should be substituted by ‘here the challenge is’. The point is not to state that something is difficult or hard but rather to underline the challenges uderlying the achievement of a certain control goal. 
  2. Could the abbreviations table be anticipated? It would be much more useful at the beginning of the manuscript rather than at the end.

Reviewer 3 Report

  1. The Section 1 - Introduction:
    • According to the survey, the topic of Circular formation seems to be extensively studied and there will not be many possible extensions. It would be best if the authors can give an outlook on the topic.
    • The review paper cites a lot of papers, some are not directly related to the topic of "circular formation". Although a survey paper usually aims to give a broad view of the discussed topic, citing too many papers will distract the readers from the important ones. 
    • The statement "the goal of the circular formation control is to drive group of agents to follow a circular trajectory/circular motion" may exclude many works in this field. The reviewer would like to suggest "circular formation control is formation control where the interactions between agents are characterized by a cycle/directed cycle graphs (with possibly the addition of a center node)." Although a large portion of works in circular formation control solves the formation rotation, the rotation is a feature which often be included with circular formation. Rotating formation can be solved with other kinds of interaction graphs, e.g., please check 10.1016/j.sysconle.2010.06.015
    • In page 2, the reviewer feels hard to follow the classification of existing results in circular formation control. The paper uses different classifications such as history (early works to more recent paper), agent's model (1st order-, 2nd order- dynamics, constrained model, constraints in communications or goals,...), methods, and then results and applications. I would suggest the authors to compress the introduction so that it briefly introduces the main aim of the paper: "techniques" in the circular formation control. Also, in the introduction, applications of cyclic pursuit (Section 6) may be presented to emphasize the importance of the topic.
  2. The problem formulation:  This model (1) seems follows the survey on formation control of Oh & Park & Ahn. Is it meaningful to include (1) without specifying the goal in mathematical notation? It would help if the control objective be written as a set of which the output of the system (1) should converge to after t \to \infty. The authors may follow the definition in 10.1109/CDC.2008.4739000
  3. In the notation in Table 1: the reviewer cannot understand the definition of class S function. Please check it again. Also, it seems that the table can be removed and the notations can be defined when they are used in the first time.
  4. Subsection 2.2:
    • Again, the classifications of CFP into CCM and Circumnavigation are respectively formation control with cycle/directed cycle or cycle/directed cycle with an additional center node.
    • CFP is further divided into even and arbitrary distribution. In the language of formation control, the offsets between agents are ussually be considered to be arbitrarily defined. It would help if the authors specify the main difficulties in making the distribution arbitrary in compare with equally spaced case?
    • In a sense, the sentence "Based on the aforementioned observations, the CFP methods  are classified as: Artificial potential method [115,116], cyclic pursuit method [117,118], leader-follower method [75,111], behavior-based method [119,120], and virtual structure method [121, 122]. The comparison of mentioned methodologies is summarized in Table 2. Note that above methods can be used together or separately to solve the CFP [103,123]" is not very logical with the aboved classifications. Table 2 is misleading since "circular formation" implies no agent acts as a leader, the interaction graph should be a cycle. In this sense, only the cyclic method in Table 2 is relevant. So, the reviewer suggests omitting Table 2 and focusing on "cyclic pursuit".
  5. The categorization by sensing variables in Section 2.3 is quite clear.
    • The authors seem to missed the relative position or displacement-based measurement approach since it was discussed in cyclic pursuit method earlier in subsection 2.2. So, some editing or combining Table 2 into this part will make the survey more logical.
    • The bearing measurements subsection: the bearing angle of agent a wrt agent b is an angle, so it may be better to specify which axis agent a is refered to to determine \theta_{ab}. Also, the survey seems not to consider the cyclic formation control in 3D of higher dimensional space. In several works, the bearing vectors or line-of-sight are more general and being used instead of angles. Please check, e.g., 10.23919/ECC.2018.8550111,  10.1016/j.sysconle.2013.10.003 
    • The RSS also gives range/distance information. May it be combined with range measurement subsection? 
  6. Section 3: CFP Methods
    • The control law (3) is introduced but it still quite brief for readers to understand. Maybe a sketch of analysis will make this main section more meaningful. In the  cyclic pursuit problem, the spectrum of the Laplacian plays an important role, which would be mentioned in the section.
    • Similarly, the readers will be curious on specific techniques used in the target circular formation tracking control. May the control law (4) be explicitly given in a very simple form?
  7. Section 5: 
    • It is not clear what criteria are used for categorizing some controller to be an advanced research schemes? The reviewer would suggest changing the title of this section to "Other research schemes". 
    • Event-trigger is a control method; "formation with constraints" is a large topic since constraints can be understood as constraints in agent's model (nonholonomic, underactuated models, saturation in input/output/state...) or constraints in network such that communication, sensing,...; and "source seeking" is a kind of application. So it will be messing to group all three topics into one.
  8. Section 6: Figure 8 seems to have overlaps between applications. Formation control has both academic interests and application interests while circular formation is a small branch, which used for very specific application. It seems that removing Fig. 8 will not lessen the contribution of the paper.
  9. Section 7: a more detail discussions on possible future research directions will be interested to the readers. 

Reviewer 4 Report

A class of circular formation of Multi-Agent Systems (MASs) are brought in this paper. The investigated topic is interesting and the mathematical explanation of the proposed solutions is also nice. Generally, the paper is well written and organized. However, some indistinct expressions abate the readability of this paper. Thus, I will recommend several clarifications before acceptance of this paper which is listed below:

  1. The motivation on the study should be further emphasized, particularly; the main advantages of the results in the paper comparing with others should be clearly demonstrated. 
  2. In Table 2, the purpose of each of the two examples needs to be explained. In addition,  more discussion needs to be added to highlight the advantages of the proposed design method. 
  3.  Some spelling errors and mistakes can be found in the paper. Authors must read text carefully eliminating those errors and mistakes.
  4.   What can we learn from this paper? One Remark should be
    given to show the difference and novelty.

  5. Another control solution is a hot topic recently and has been applied in synchronization of the dynamical Systems, such as "Sampled-data synchronization of delayed multi-agent networks and its application to coupled circuit; Non-fragile synchronisation of mixed delayed neural networks with randomly occurring controller gain fluctuations; Extended dissipativity and non-fragile synchronization for recurrent neural networks with multiple time-varying delays via sampled-data control;  Dynamical analysis and sampled-data stabilization of memristor-based Chua’s circuits".
  6.   It seems that the technique of this paper is well-known. The authors must clearly show the difference and improvements in comparison with the existing results in the view of technique analysis. 
  7.   Symbol format should be more standardized. 
  8.   The labels of some equations in the main results section are missing. 
  9.   The authors should provide some future topics in the conclusion section. 
  10.   The format in the reference article needs to be uniform such that the
    source of the cited reference requires a uniform format. Please check them.

Round 2

Reviewer 3 Report

I think the paper has been improved. But some references are still not very relevant (there are too many indeed). On the other hand, several works can be included, e.g., 10.1109/ACC.2013.6580804 and 10.1109/TAC.2016.2631302. Other than this point, I have no further comment.

Author Response

Manuscript ID:  electronics-1444735

Title:  A Survey on Techniques in the Circular Formation of Multi-Agent Systems

Authors:  Hamida Litimein, Zhen-You Huang and Ameer Hamza*

Responses to Reviewer 3

Thank you for considering our manuscript for review and giving us the opportunity for revision. 0We appreciate your time to process this manuscript and give review comments immediately. The details of revisions and responses to reviewers’ comments are listed as following pages.

Comment. I think the paper has been improved. But some references are still not very relevant (there are too many indeed). On the other hand, several works can be included, e.g., 10.1109/ACC.2013.6580804 and 10.1109/TAC.2016.2631302. Other than this point, I have no further comment.

Author response: Thanks for your constructive comment. According to your suggestions, we removed 40 references as follow:

[2], [4], [7], [10], [11], [12], [13], [15], [18], [20], [21], [36], [55] ,[117], [119], [123], [124], [127], [132], [136], [141], [144], [143], [145], [149], [155], [160], [172] [183], [184], [226], [258], [203], [221], [265], [270], [285], [294], [245],[280].

The reduction of references is done on the basis of similar work in a particular section, the latest reference (article) has been considered while the earliest work has been removed.

According to your suggestions, we added the recommended references in section 3, subsection “Circular formation based cyclic pursuit strategy “as follows:

A generalization of the linear cyclic pursuit strategy with heterogeneous gains is studied in [66] to lead a group of agents to rendezvous at some desired point. The authors showed that by choosing proper gains, the agents could achieve rendezvous at some desired point.

Further, by selecting proper deviations on agents, [129] expands the reachability set presented in [66] to include points that are not reachable using generalized cyclic pursuit strategy.

The necessary and sufficient conditions on the consensus of heterogeneous cyclic pursuit strategy in discrete time are established in [130].

Mukherjee, D.; Ghose, D. On Synchronous and Asynchronous Discrete Time Heterogeneous Cyclic Pursuit. IEEE Trans. Autom. Control  2017, 62, 5248-5253.

Mukherjee, D.; Ghose, D. Generalization of deviated linear cyclic pursuit. In Proceedings of the 2013 American Control Conference, Washington, DC, USA, 17-19 June 2013; pp. 6163- 6168.

Reviewer 4 Report

In a general way, most of my comments were answered by the authors. The paper can be accepted in its present form.

Author Response

Manuscript ID:  electronics-1444735

Title:  A Survey on Techniques in the Circular Formation of Multi-Agent Systems

Authors:  Hamida Litimein, Zhen-You Huang and Ameer Hamza*

Responses to Reviewer 4

Comment: In a general way, most of my comments were answered by the authors. The paper can be accepted in its present form.

Author response: Thank you for considering our manuscript for review and giving us the opportunity for revision. We appreciate your time to process this manuscript and give review comments immediately. 

We have carefully read the manuscript and improved the writing style as per the suggestions as well.